🔓 | **Open Peer Review** | Environmental Microbiology | Research Article

# Molecular and evolutionary basis of O-antigenic polysaccharide-driven phage sensitivity in environmental pseudomonads

Jordan Vacheron,[1] Clara M. Heiman,[1] Julian R. Garneau,[1] Peter Kupferschmied,[1] Ronnie de Jonge,[2] Daniel Garrido-Sanz,[1] Christoph Keel[1]

**ABSTRACT** *Pseudomonas protegens* CHA0, a bacterial strain able to suppress plant pathogens as well as efficiently kill lepidopteran pest insects, has been studied as a biocontrol agent to prevent ensuing agricultural damage. However, the success of this method is dependent on efficient plant colonization by the bacterial inoculant, while it faces competition from the resident microbiota as well as predators such as bacteriophages. One of these naturally occurring phages, ΦGP100, was found to drastically reduce the abundance of CHA0 once inoculated into plant microcosms, resulting in the loss of plant protection effect against a phytopathogen. Here, we investigated the molecular determinants implicated in the interaction between CHA0 and the phage ΦGP100 using a high-density transposon-sequencing approach. We show that lipopolysaccharide cell surface decorations, specifically the longer OBC3-type O-antigenic polysaccharide (O-PS, O-antigen) of the two dominant O-PS of CHA0, are essential for the attachment and infection of ΦGP100. Moreover, when exploring the distribution of the OBC3 cluster in bacterial genomes, we identified several parts of this gene cluster that are conserved in phylogenetically distant bacteria. Through heterologous complementation, we integrated an OBC3-type gene copy from a phylogenetically distant bacterium and were able to restore the phage sensitivity of a CHA0 mutant which lacked the ability to form long O-PS. Finally, we evidence that the OBC3 gene cluster of CHA0 displays a high genomic plasticity and likely underwent several horizontal acquisitions and genomic rearrangements. Collectively, this study underlines the complexity of phage-bacteria interactions and the multifunctional aspect of bacterial cell surface decorations.

**IMPORTANCE** The application of plant-beneficial microorganisms to protect crop plants is a promising alternative to the usage of chemicals. However, biocontrol research often faces difficulties in implementing this approach due to the inconsistency of the bacterial inoculant to establish itself within the root microbiome. Beneficial bacterial inoculants can be decimated by the presence of their natural predators, notably bacteriophages (also called phages). Thus, it is important to gain knowledge regarding the mechanisms behind phage-bacteria interactions to overcome this challenge. Here, we evidence that the major long O-antigenic polysaccharide (O-PS, O-antigen) of the widely used model plant-beneficial bacterium *Pseudomonas protegens* CHA0 is the receptor of its natural predator, the phage ΦGP100. We examined the distribution of the gene cluster directing the synthesis of this O-PS and identified signatures of horizontal gene acquisitions. Altogether, our study highlights the importance of bacterial cell surface structure variation in the complex interplay between phages and their *Pseudomonas* hosts.

**KEYWORDS** bacteriophages, environmental microbiology, molecular biology, *Pseudomonas*, virus-host interactions

Address correspondence to Jordan Vacheron, jordan.vacheron@unil.ch, or Christoph Keel, christoph.keel@unil.ch.

The authors declare no conflict of interest.

See the funding table on p. 16.

The use of plant-beneficial microorganisms in agriculture is a promising alternative to pesticides for controlling plant pathogens and pests (1). Nonetheless, this method is dependent on the successful establishment of a microbial inoculant within the plant environment where it faces competition and predation from the resident plant microbiota.

*Pseudomonas protegens* bacteria, which are represented by the type and model strain CHA0 (2), are well-known plant root colonizers that can provide various plant-beneficial functions, which include, amongst others, suppression of diseases caused by phytopathogenic fungi and oomycetes, nutrient mobilization, and stimulation of plant growth and defenses (3). *P. protegens* bacteria are also capable of efficiently infecting and killing lepidopteran pest insects that cause serious damage to agricultural crops (1, 4–6). All these plant-beneficial traits make these bacteria promising candidates in assays aimed at improving plant health and performance in field conditions (7, 8). However, the transition from a controlled laboratory environment to the field constitutes a considerable leap into a system influenced by a plethora of biotic and abiotic factors. Notably, the resident root microbiota can be perceived as a biological barrier (9), which can interfere with the establishment of the bacterial inoculant. Indeed, inoculants will face competition from other members of the host microbiota for nutrients and space. Moreover, the root microbiota also harbors predators of bacteria, which will strongly influence the fate of the inoculant, and thereby may reduce its plant-beneficial activities. Amongst these predators, grazers such as protozoa and nematodes (10, 11) as well as bacteriophages (12) are known to shape the structure of the bacterial community within the root microbiota.

Bacteriophages, or briefly phages, are a type of virus that infects bacteria and use this host as a cellular replication factory to maintain their viral population within a specific environment. Phages exhibit an impressive diversity, reflecting the variety of the bacterial microbiota and the environments they interact with [for a review see reference (13)]. They play crucial roles in different ecosystems by controlling the abundance and diversity of bacterial populations as well as maintaining a dynamic gene flow within bacterial communities through horizontal gene transfers (14–17). Moreover, the interest in phages as biocontrol agents is rising and provides promising results for the protection of plants against bacterial diseases (18, 19). However, naturally occurring phages may also target bacterial inoculants used to benefit agricultural crops. Indeed, a lytic bacteriophage that targets the plant-beneficial *P. protegens* strain CHA0 was isolated from the rhizosphere of cucumber plants (20, 21). The presence of this phage, named ΦGP100, drastically reduced the abundance of CHA0 in the rhizosphere, resulting in the loss of the plant protection effect against a root pathogenic oomycete (20).

Phages generally target specific cell surface structures for initial attachment to their host cell, such as exopolysaccharides, pilus or flagellum proteins as well as lipopolysaccharides (LPS) for Gram-negative bacteria (22–24). LPS are the principal components of the outer membrane of Gram-negative bacteria and generally are composed of three parts, the lipid A, the core oligosaccharide (core-OS), and the O-antigenic polysaccharide (O-PS, O-antigen) (25–27). The lipid A, which is anchored in the outer membrane, is bound to the core-OS made of different sugar residues. The lipid A-core-OS constitutes the conserved structure to which the highly variable O-PS is ligated, consisting of an assembly of repetitive glycans that are subject to various modifications. Due to their exposure at the cell surface, the O-PS components are of high importance in bacteria-host and intermicrobial interactions and a virulence factor in many pathogenic bacteria (25–27). O-PS significantly contributes to the insect pathogenicity and competitiveness of *P. protegens* strains (28, 29). *P. protegens* CHA0 harbors four O-PS gene clusters specifying the formation of OSA, OBC1, OBC2, and OBC3, of which OSA and OBC3, a short and a long O-PS respectively, constitute the dominant LPS decorations of the strain (28).

To better understand the interaction between a root-colonizing beneficial bacterial inoculant (*P. protegens* CHA0) and its naturally occurring phage predator (ΦGP100), we

identified potential bacterial genetic determinants involved in the susceptibility toward the phage ΦGP100, using a high-density transposon-sequencing (Tn-seq) approach. Through a reverse genetics approach, we provide evidence that the OBC3-type O-PS, which constitutes the major long O-PS of *P. protegens* CHA0, is the receptor of the lytic phage ΦGP100. We then examined the distribution of the OBC3 gene cluster in bacterial genomes and identified several parts of this gene cluster, which are conserved in phylogenetically distant bacteria. We were able to restore the phage sensitivity of a CHA0 mutant depleted for the formation of the long O-PS by heterologous complementation, integrating a copy of a related OBC3-type gene from one of these phylogenetically distant bacteria. Finally, we evidence that the OBC3 gene cluster of CHA0 displays a high genomic plasticity as well as signatures of sequential acquisitions by horizontal gene transfer.

## RESULTS AND DISCUSSION

### Identification of genes required for the ΦGP100 phage resistance using a transposon-sequencing approach

To identify the genes contributing to the sensitivity of *P. protegens* CHA0 toward the phage ΦGP100, we generated a saturated transposon mutant library using the Tn*5* transposon carried by the pRL27 plasmid [Table S1 (30)]. We obtained approximately 500,000 transposon insertions distributed throughout the genome of *P. protegens* CHA0 (i.e., around 70 transposon insertions every 1,000 bp) (Fig. S1A; Table S2). More than 93% of the genes were affected by the insertion of the transposon with an average between 67 and 84 transposon hits per gene (Table S2). The Tn-seq results from the different conditions were highly reproducible with a Pearson correlation coefficient of 0.98 (Fig. S2).

To detect a wide range of genes involved in the sensitivity of CHA0 to the phage ΦGP100, we exposed the Tn-mutant library to a gradually increasing concentration of phage particles. After an overnight exposure, we recovered the enriched Tn-mutants that had incorporated the Tn5 transposon in crucial genes responsible for the sensitivity of this strain toward the phage ΦGP100 and proceeded with DNA extraction and sequencing to identify these genes (Fig. 1A; Fig. S1B through D). For the three different multiplicity of infection tested (MOI = 1; MOI = 10; MOI = 100), we obtained 3,467, 4,029, and 4,074 genes, respectively, which were significantly less represented in the Tn-mutant library following the exposure to the phage ΦGP100 (Fig. 1B; Fig. S3). The genes identified encompass essential genes for the development and growth of the bacterial host, as well as genes for which the disruption by the transposon might increase the sensitivity of the bacterium to the phage. Applying a gradually increasing concentration of phages allowed a deeper look into the phage infection process by focusing on bacterial genes potentially needed for phage replication inside the cell. Indeed, by comparing the different phage concentrations, i.e., an MOI of 100 vs 10 and an MOI of 100 vs 1, we identified 30 and 13 genes, respectively, related to the metabolism of the bacterial cell and the transcription machinery, including transcriptional and translational regulators (Fig. S4; Data set 1). Conversely, a total of 385, 210, and 232 genes were detected as potentially involved in the bacterial sensitivity toward the phage at an MOI of 1, 10, and 100, respectively (Fig. 1B; Fig. S3). Globally, Tn-mutants that were the most enriched following phage infections were highly similar for all the different MOI applied (Fig. S1). The exposure time of the Tn-mutant library (10 h) might explain these similarities between the different phage concentrations applied. Although most of the identified genes were assigned as an unknown function according to the eggNOG functional annotation, 13% were associated with the cell wall and membrane biosynthesis category (Fig. 1B; Fig. S3). Most of the genes from this EggNOG functional category participate in different steps of the biosynthesis of LPS (Data set 1). Indeed, the three most abundant Tn-mutant categories (Fig. 1A; Data set 1) possessed transposon insertions in the *lapA* gene, encoding an adhesin involved in biofilm attachment (31), in *galU,* required to form an operational LPS core-oligosaccharide in *Pseudomonas*

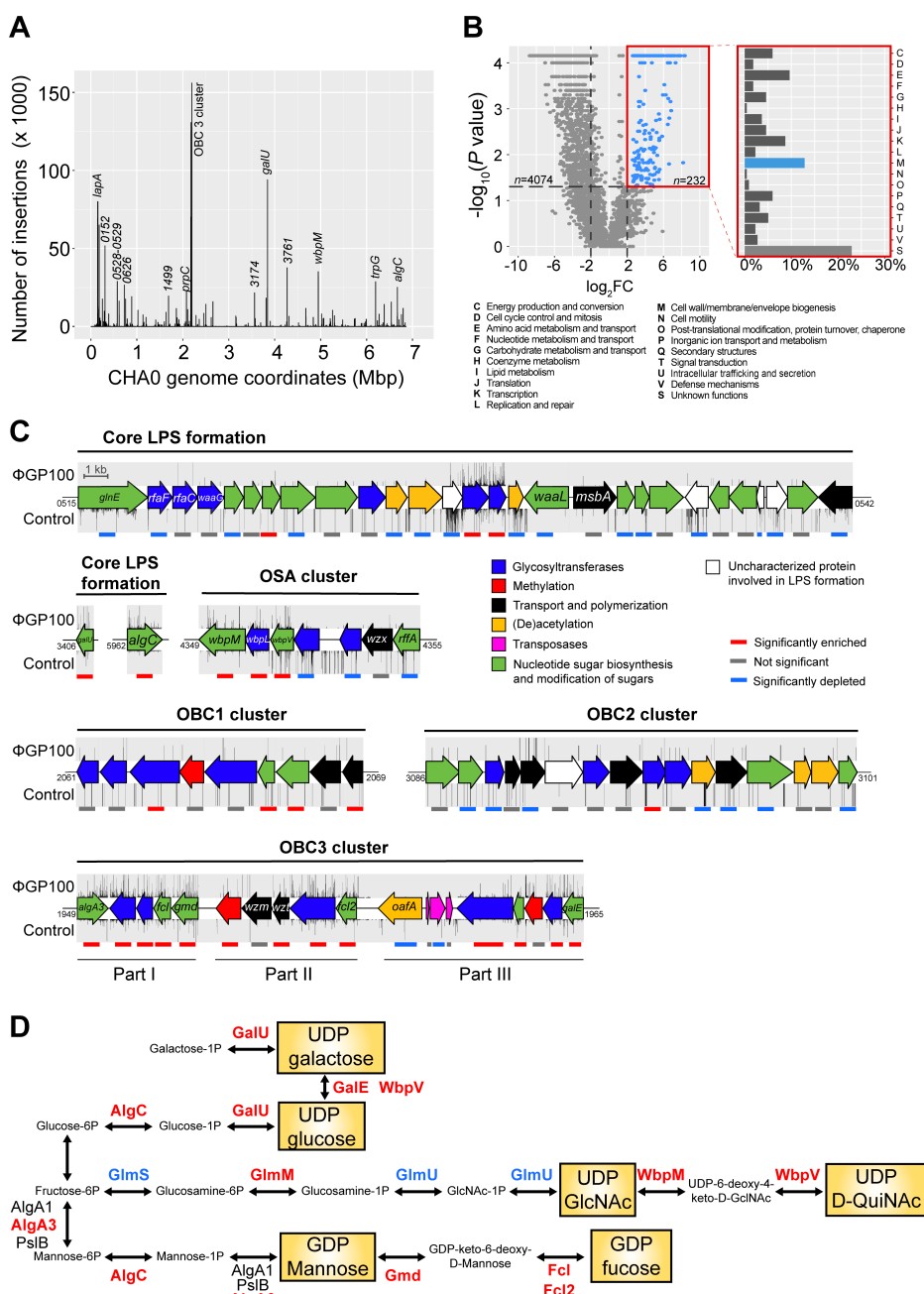

**FIG 1** Tn-seq analysis specifies LPS components of *Pseudomonas protegens* CHA0 as the main receptors of the phage ΦGP100. (A) Distribution of the transposon insertions across the genome of *P. protegens* CHA0 in the presence of the phage ΦGP100 at an MOI of 100. The results for all MOI levels assessed in this study are available in Fig. S1. (B) Volcano plot highlighting the enriched 232 candidate genes involved in the phage ΦGP100 resistance and their COGs' assignation. (C) Visualization of the transposon enrichment in the different gene clusters involved in the LPS formation of CHA0 upon exposure to ΦGP100. The density of the transposon insertions is indicated above (condition with the phage) or below (control condition) the genes. The colored lines below the gene clusters refer to the Tn-seq results: gray bars refer to a non-significant involvement of the corresponding gene in the phage sensitivity (−2 < log2 FC < 2 and *P* > 0.05); red and blue lines refer to a significantly enriched (log2FC > 2 and *P* < 0.05) or depleted (log2FC < −2 and *P* < 0.05) number of transposon insertions, respectively, in the corresponding genes. Numbers at the start or the end of the respective clusters correspond to the gene locus tags for *P. protegens* CHA0 (prefix PPRCHA0_…). (D) Biosynthetic pathways of the nucleotide sugars involved in the LPS formation, which are important for the sensitivity of CHA0 to ΦGP100. Enzymes written in red or blue display, respectively, a significantly higher or lower number of Tn insertions in their respective coding genes when CHA0 is exposed to ΦGP100. Enzymes written in black were not significantly highlighted in the Tn-seq analysis.

*aeruginosa* (32, 33) or in the OBC3 gene cluster, specifying the formation of the major long O-PS in *P. protegens* CHA0 (28).

## The long O-PS of CHA0 is mandatory for the infection by ΦGP100

Following the results from the Tn-seq experiment, we decided to better characterize the role of the LPS of *P. protegens* CHA0 in the interaction with the phage ΦGP100. The Tn-seq experiment highlighted the importance of a complete and operational core-OS for the phage infection process (Fig. 1C). Upon ΦGP100 phage exposure, Tn insertions were identified in several core LPS genes, notably in *galU* and *algC* (Fig. 1C). The gene *galU* encodes a UTP-glucose-1P uridylyltransferase, which allows the production of UDP-galactose and UDP-glucose (Fig. 1D) that are essential sugar residues composing the core-OS in *P. aeruginosa* (32). Indeed, a *galU* mutant of *P. aeruginosa* was no longer able to bind O-PS as the core-OS appeared truncated (33). The gene *algC* encodes a phosphomannomutase/phosphoglucomutase, a key enzyme that catalyzes the transformation of glucose-6P into glucose-1P and mannose-6P to mannose-1P (Fig. 1D), which also are essential components of the core-OS (34). Indeed, an *algC* mutant of *P. aeruginosa* harbored a truncated core-OS unable to attach any O-PS (34), resulting in resistance toward two different phages (35). Other genes involved in the core-OS formation, such as *wbpM* and *wbpL,* were identified by the Tn-seq experiment (Fig. 1C). These two genes are located in the OSA cluster in the genomes of *P. aeruginosa* and *P. protegens* CHA0 and are conserved among various other *Pseudomonas* genomes (28). The glycosyltransferase WbpL initiates the synthesis of the O-PS by transferring an N-acetyl sugar to the lipid carrier in the cytoplasm of the bacterium (36–38). The deletion of *wbpL* in CHA0 and in other *Pseudomonas* results in strains unable to bind O-PS (28, 39). Finally, genes involved in the metabolism of amino acids and raw sugars were impacted, likely not only due to their essential role in bacterial growth but also as they supply the raw sugar subunits for the core-OS and the O-PS biosynthesis (Fig. 1D).

*P. protegens* CHA0 harbors four O-PS gene clusters specifying the formation of OSA, OBC1, OBC2, and OBC3, of which OSA and OBC3, a short and a long O-PS, respectively, constitute the dominant LPS decorations of the strain (28). In the present study, only Tn-insertions within the OBC3 gene cluster were enriched upon exposure to the phage ΦGP100 (Fig. 1A and C). Then, 13 out of the 19 genes found in the OBC3 gene cluster showed a significantly higher number of transposon insertions compared to the control condition (Fig. 1C). To confirm the Tn-seq results, we tested deletion mutants impaired in the production of the different O-PS and confronted them with the phage (Fig. 2A; Fig. S5). A *wbpL* mutant of CHA0, which lacks both OSA and OBC3 (28), was fully resistant to the phage ΦGP100 (Fig. 2A). Neither the OBC1 nor the OBC2 mutants showed a different sensitivity pattern toward the phage compared to the wild-type CHA0 (Fig. 2A), supporting the Tn-seq data. However, the OBC3 mutant, which lacks a significant part of the OBC3 cluster (part I and part II of the cluster, Fig. 1C), became fully resistant to the phage (Fig. 2A), strongly suggesting that the ΦGP100 phage targets the long O-PS displayed at the cell surface of CHA0. Phage resistance could also be achieved by deleting individual genes of the OBC3 cluster, including *fcl*, *fcl2*, PPRCHA0_1957, PPRCHA0_1962, and *galE* (Fig. S5). Although most of the genes belonging to the OBC3 gene cluster encode important enzymatic functions necessary for the production of the long O-PS, the gene *oafA* encoding an O-antigen acetylase (28, 40) was not detected as an important gene involved in the sensitivity of CHA0 toward the phage ΦGP100 by the Tn-seq analysis (Fig. 1C). Furthermore, a deletion mutant of *oafA* was still sensitive to the phage (Fig. S5), further corroborating the Tn-Seq data. Even though the acetylation of the long O-PS does not seem to play a role during the infection of *P. protegens* CHA0 by the phage ΦGP100, it could still be important for the ecology of this bacterium, for example, during the interaction with its eukaryote hosts such as plants and pest insects as suggested by a study on the experimental evolution of bacterial root colonization traits (40).

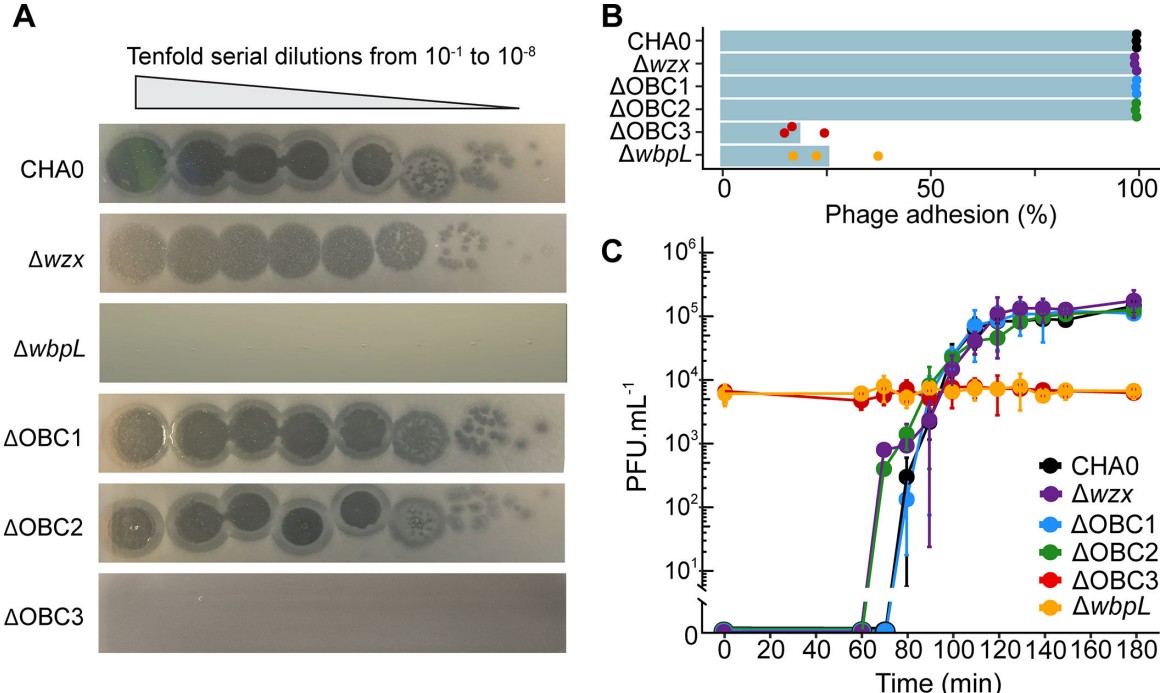

**FIG 2** The long O-PS of CHA0 is mandatory for phage infection. (A) Spot assays of ΦGP100 onto a bacterial lawn of wild-type CHA0 or mutants impaired in the production of different O-PS. The wild-type CHA0 produces two major O-PS, the short OSA and the long OBC3 (28). The ΔOBC3 mutant lacks the OBC3 O-PS, whereas the Δ*wzx* mutant does not possess an OSA O-PS. The Δ*wbpL* mutant lacks both O-PS. OBC1 and OBC2 make no apparent contribution to O-PS formation (28). (B) The number of new phages released following infection of CHA0 wild type and the different O-PS mutants of CHA0. (C) Phage adsorption onto the cell surface of CHA0 wild type and the different O-PS mutants of CHA0.

Interestingly, the plaques formed by the phage ΦGP100 when deposited onto CHA0 during spot assays exhibited a clear center indicative of a complete bacterial lysis, surrounded by a turbid halo (Fig. 2A). The deletion of the flippase-encoding gene *wzx*, which leads to the loss of the short OSA-type O-PS but maintains the long OBC3 O-PS, produced a mutant that did not display a clear center, but the entire plaques remained turbid (Fig. 2A). The presence of such turbid zones is typical of phages exhibiting a depolymerase activity (41), which could be, in the case of the phage ΦGP100, attributed to the degradation of the long OBC3 O-PS of CHA0. The absence of the clear center when the phage infects Δ*wzx* was not related to either a lower phage adhesion on the cell surface (Fig. 2B) or a decrease of the phage infectivity (Fig. 2C). The depolymerase activity is frequently observed in phages of the *Podoviridae* family (42, 43). Although we previously thought that the ΦGP100 phage belonged to the *Podoviridae* family considering its shape under electron microscope (20), the new genomic analysis we performed here assigned it to the *Zobellviridae* family (44), displaying, however, low similarity with other closely related phages from this family (Fig. 3A; Table S3). Furthermore, we performed a new annotation of the phage genome (Fig. 3B; Table S4) to detect potential genes encoding depolymerases that could be responsible for the halo formation (Fig. 2A). We identified a gene predicted to encode a protein containing a pectate lyase domain (Fig. 3B; Fig. S6). The protein structure of this pectate lyase was predicted and displays structural similarities with a pectate lyase encoded in the genome of the phage P1 (Fig. S6) where the depolymerase activity had been demonstrated to be responsible for the formation of a halo when the phage infects *Acinetobacter* strains (45).

Based on these results, we propose the following model for the infection of CHA0 by the phage ΦGP100 (Fig. 3C). The phage recognizes the long O-PS (OBC3) at the surface of CHA0 (and other *P. protegens* group strains displaying OBC3, see below) as a primary receptor for the infection process. Then, we suggest that the phage deploys the

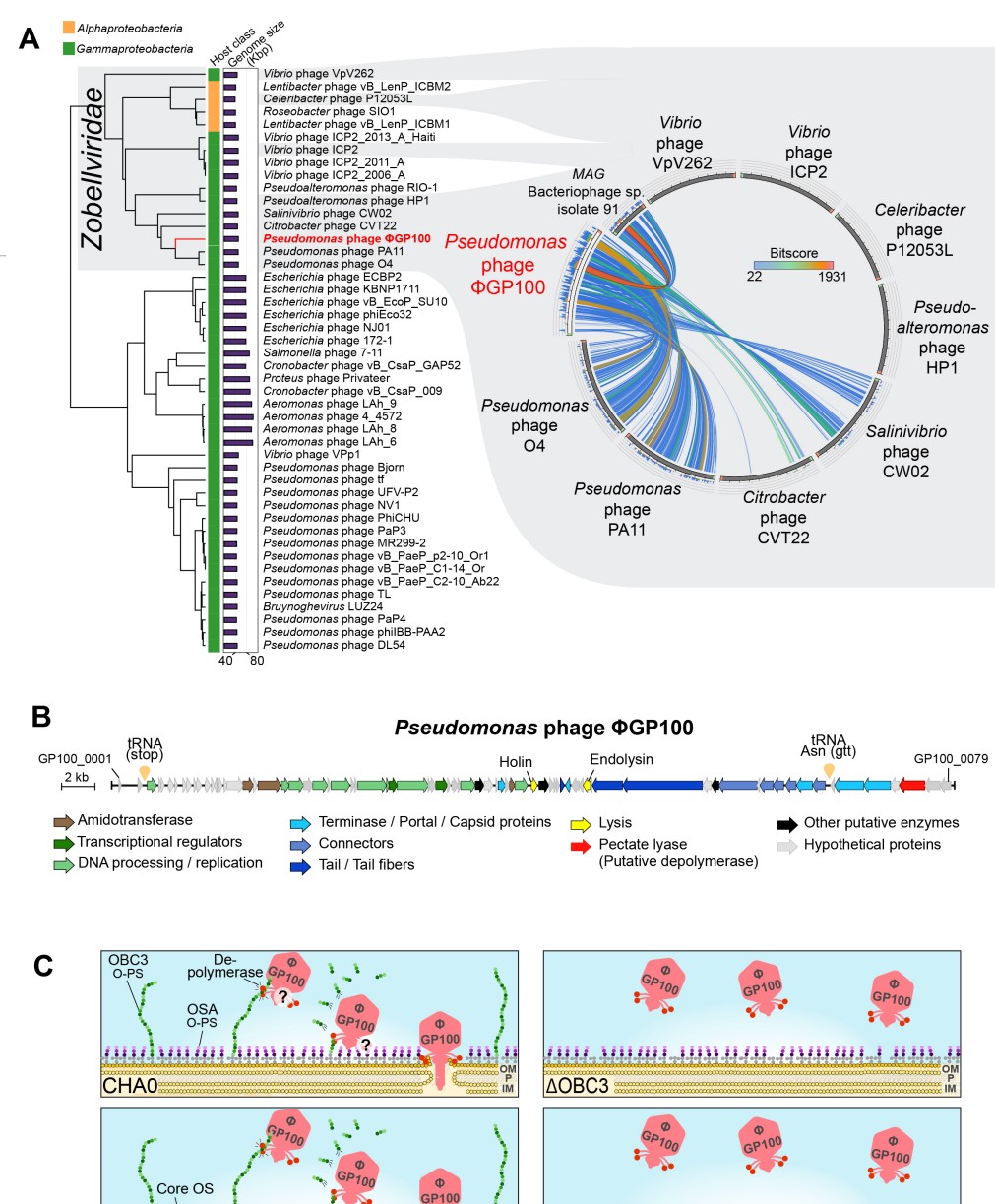

**FIG 3** Characterization of the phage ΦGP100 and potential mode of action toward CHA0. (A) Taxonomy assignment of the phage ΦGP100 into the *Zobellviridae* family. The proteomic tree was generated on the ViPTree online platform (46). The host category as well as the sizes of the different phage genomes are displayed with colored vertical and horizontal bars, respectively. The genomic similarity between the phage ΦGP100 and other closely related phages from the *Zobellviridae* family is visualized alongside the proteomic tree. Colors reflect the similarity levels related to the values of the maximum hit scores (Bitscore). The list of genomes used in this panel is available in Table S3. (B) Update of the phage annotation with the identification of the pectate lyase that is potentially involved in the depolymerization of the long OBC3-type O-PS of CHA0. Genes are color-coded according to their protein functions. The detailed functional annotations of the ΦGP100 phage are provided in Table S4. (C) Schematic representation of the proposed infection process of the phage ΦGP100 according to the sensitivity profiles obtained for the different O-PS mutants. ΦGP100 targets the long OBC3-type O-PS at the surface of CHA0 cells and displays a depolymerization activity that potentially degrades this long O-PS (see plaques for the *wzx* mutant in Fig. 2A). The localization of the depolymerase on the phage as well as its interaction with the short O-PS (OSA type) still need to be investigated, which is the reason why they are represented with interrogation points. OM: outer membrane; P: peptidoglycan; and IM: inner membrane.

depolymerase activity to degrade the long O-PS to reach the bacterial outer membrane and injects its genetic material inside the bacterial cell (see plaques for the *wzx* mutant in Fig. 2A). However, the interaction between this depolymerase with the short O-PS (OSA type) remains to be investigated as well as the localization of the depolymerase on the phage (i.e., on the tail fibers, tail spike, or the neck).

## Distribution of the OBC3 gene cluster and host range of the ΦGP100 phage

Next, we examined the distribution of the OBC3 gene cluster as well as the OSA gene cluster within bacterial genomes from the NCBI database to see whether the presence of the OBC3 gene cluster or parts of it can be linked to the phage sensitivity (Fig. 4). We split this gene cluster into three parts according to the arrangement of the genes within the cluster (Fig. 1C). We looked at the presence/absence of the three parts composing the OBC3 gene cluster by Blastp, applying a threshold of 60% of amino acid identity on at least 70% of the amino acid sequence length. In parallel, we determined the activity spectrum of the phage ΦGP100 against different *Pseudomonas* strains as well as against bacteria where we detected the entire or parts of the OBC3 gene cluster.

Half of the strains belonging to the *P. protegens* species investigated in this study (CHA0, Pf1, PGNR1, BRIP, and Cab57) possess the OSA gene cluster and the OBC3 gene cluster. Moreover, the OBC3 gene cluster is located in the same genomic region of these bacterial genomes, i.e., between a gene encoding an aerotaxis receptor (*aer_2*) and a gene encoding a CPBP family intramembrane metalloprotease (Fig. 4). Although these bacteria harbor the OSA gene cluster along with the OBC3 gene cluster, BRIP and Cab57 exhibited a reduced sensitivity to the phage ΦGP100 compared to CHA0 (Fig. 4; Fig. S7). Since different strategies are deployed by bacteria to evade phage infection (48), the difference in phage sensitivity between CHA0 and BRIP/Cab57, despite harboring the same OBC3 O-PS (28), may be based on the presence of intracellular anti-phage defense systems such as CRISPR-cas systems (48), which could limit the infection by the phage ΦGP100 (48). CMR5c is also less sensitive to the phage compared to CHA0 (Fig. 4; Fig. S7). However, in this strain, only part III of the OBC3 gene cluster was detected at the same chromosomal locus as in CHA0, while the two other parts are located in another genomic region and are not entirely complete (Fig. 4). On the contrary, we detected parts II and III of the OBC3 cluster of *Pseudomonas* sp. R76 in the same genomic region as in CHA0 (Fig. 4). Furthermore, several genes from the different parts of the OBC3 gene cluster were identified in genetically distant bacteria, belonging to the Alphaproteobacteria (*Rhizobium etli* CFN42 and *Rhizobium leucaenae* USDA9039) or the Betaproteobacteria (*Acidovorax avenae* subsp. *avenae* AA99, *Herbaspirillum rubrisubalbicans* M1 and *Azoarcus* sp. BH72) (Fig. 4; Table S5). All these distantly related bacteria as well as *Pseudomonas* sp. R76 were resistant to the phage ΦGP100 (Fig. 4). The presence of genes from the OBC3 cluster or parts of it within the genome of distantly related bacteria raises the question of the origin of this gene cluster in the phage-sensitive *Pseudomonas* bacteria.

## Heterologous complementation of a CHA0 mutant lacking the long O-PS restores phage sensitivity

The OBC3 gene cluster of *P. protegens* CHA0 harbors several genes of part I with relatively high sequence identities (67–75%) to genes of the O-PS biosynthesis locus of *Rhizobium etli* CFN42 (Fig. 4 and 5). Three of these genes encode enzymes (a GDP-mannose 4,6-dehydratase and a GDP-L-fucose synthetase) and a glycosyltransferase (the fucosyltransferase WreE), which are sufficient for the synthesis of GDP-L-fucose from GDP-D-mannose and the covalent binding of the sugar moiety to O-PS repeat units in *R. etli* CFN42 [Fig. 5; (49)]. The homologous genes in CHA0 were at least 67% identical to the *Rhizobium* genes and seemed to also be organized as an operon, indicating that CHA0 is capable of synthesizing GDP-L-fucose (Fig. 5). As previously reported, the deletion of *fcl* (encoding a putative GDP-L-fucose synthetase) in CHA0 resulted in the loss of long OBC3-type O-PS [Fig. 5; (28)], while re-introduction of *fcl* into the genome of Δ*fcl* rescued the biosynthesis of the long O-PS (Fig. 5). Furthermore, *cis* complementation

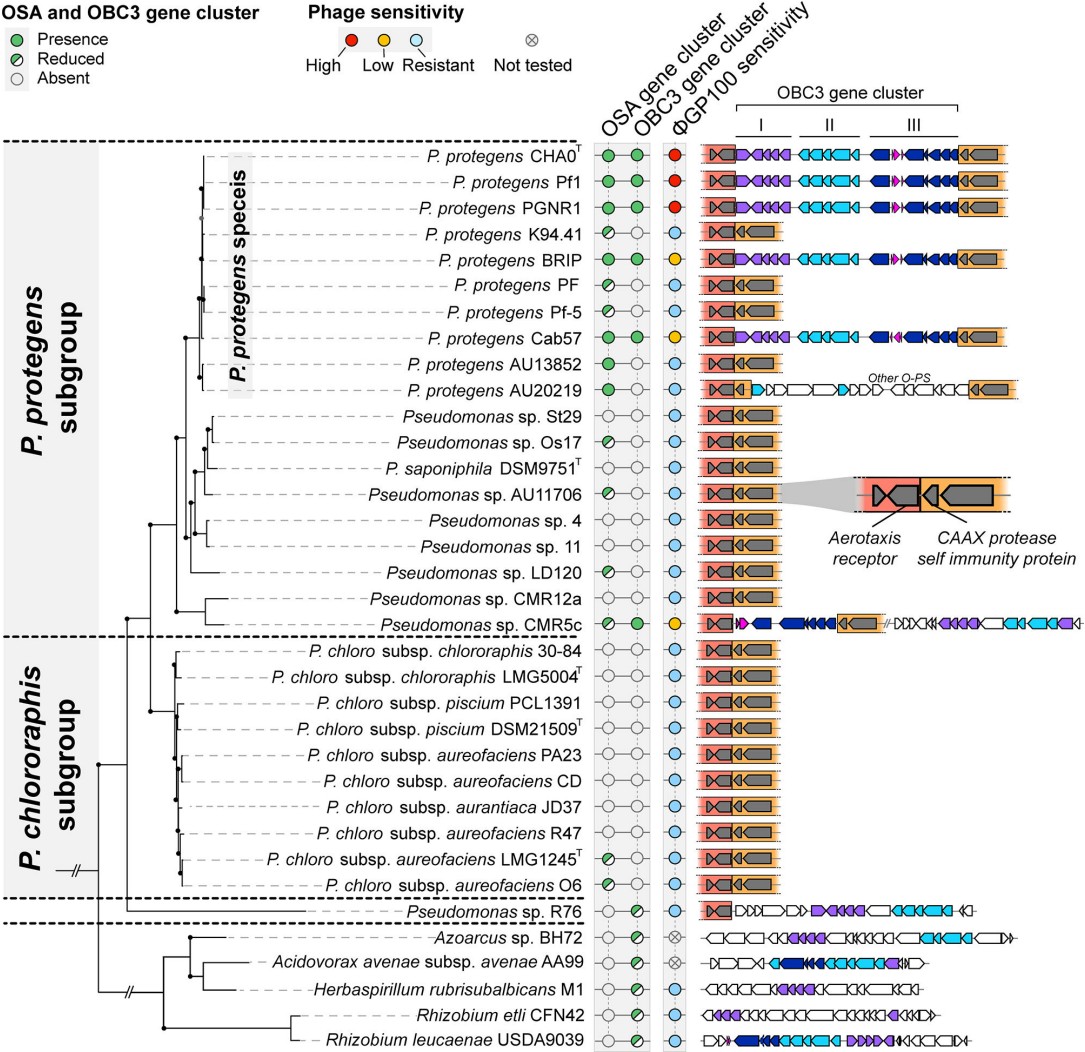

**FIG 4** Distribution of the OSA-type and OBC3-type gene clusters among bacterial genomes and phage sensitivities. The presence of the OBC3 and the OSA gene clusters was assessed by searching for orthologs of the OSA and OBC3 gene clusters of *P. protegens* CHA0 whose ensuing amino-acid sequences shared a minimum of 60% of amino acid identity on at least 70% of the amino acid sequence length. Entire or reduced versions of these two O-PS gene clusters were detected in closely related *Pseudomonas* [belonging to the *P. protegens* species *sensu stricto* (47)] or phylogenetically distant bacterial genomes. The distribution of these two O-PS as well as the sensitivity of the different strains tested are represented alongside the phylogenetic tree. The synteny of the detected OBC3 gene clusters (entire or reduced versions are represented on the right part of the figure). The OBC3 gene cluster is divided into three different parts, colored with purple, dark blue, and light blue to better visualize which parts are found in the different genomes. Parts or the entire OBC3 gene cluster were found between two specific conserved regions inside the *Pseudomonas* genomes that are underlined in dark and light orange. Genes represented in pink correspond to mobile genetic elements including transposases. The maximum-likelihood phylogenetic tree of the 34 bacterial genomes was built based on the concatenation of single-copy proteins (see the Materials and Methods section). Bootstrap values above 95% are represented with black dots.

of the same mutant strain with *fcl* from *R. etli* CFN42 similarly rescued the production of long O-PS, although this resulting O-PS appeared to have lower molecular weights compared to the wild type. Nevertheless, it can be assumed that the *fcl* gene from CHA0 encodes a GDP-L-fucose synthetase as in *R. etli* since glycosyltransferases generally display strict substrate specificity (50). The long O-PS generated by the Δ*fcl* mutant of CHA0 expressing the *fcl* gene of *R. etli* CFN42 restored the phage susceptibility of the strain (Fig. 5), indicating that acquiring genes involved in the O-PS biosynthesis from a phylogenetically distant bacterium can be functional and exploited by the recipient strain.

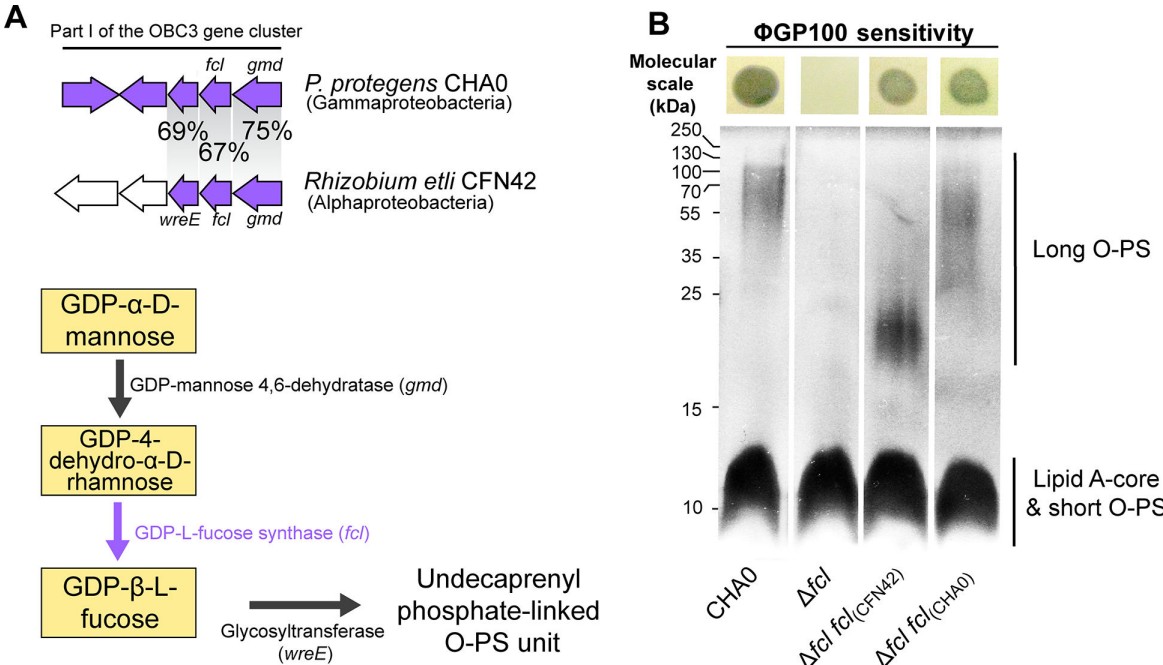

**FIG 5** The heterologous complementation of the long O-PS synthesis restores phage sensitivity. (A) Nucleotide homology between genes of part I of the OBC3 gene cluster of *Pseudomonas protegens* CHA0 and the O-PS biosynthesis locus detected in *Rhizobium etli* CFN42. Three genes of the OBC3 gene cluster show amino acid sequence identities of at least 67% with genes that are necessary for the biosynthesis of L-fucose-containing O-PS (49). GDP-L-fucose is synthesized by the conversion of GDP-D-mannose by a GDP-mannose dehydrogenase and a GDP-L-fucose synthase and can subsequently be used to assemble O-antigen units. (B) SDS-PAGE with LPS extracted from CHA0, its Δ*fcl* mutant, and *cis*- and *trans*-complemented strains. Long O-PS was lost in the Δ*fcl* mutant and (partly) rescued by complementation with *fcl* from *R. etli* CFN42 and CHA0, respectively. Each strain was tested for susceptibility to infection by phage ΦGP100 by performing a double layer assay. The corresponding pictures are shown above the gel. Clear zones indicate bacterial lysis upon phage infection.

## The OBC3 gene cluster of CHA0 possesses a high genetic plasticity and displays signatures of horizontal gene transfer acquisitions

The conserved genomic location in *Pseudomonas* genomes where the OBC3 gene cluster is integrated could be a propitious region for the acquisition of LPS gene clusters. Indeed, another O-PS gene cluster is inserted in the same conserved genomic region in strain *Pseudomonas* sp. AU20219 (Fig. 4). Moreover, a transposable element (TE) is inserted upstream of the putative O-antigen acetylase (*oafA*), within part III of the OBC3 gene cluster of *P. protegens* CHA0 (Fig. 6A). This TE is assigned as an insertion sequence (IS) belonging to the IS3 family according to the ISFinder database (51) and flanked by repetitive palindromic sequences (Fig. 6A). Interestingly, similar ISs (i.e., belonging to the IS3 family) were detected within part III of the OBC3 gene cluster in other bacterial genomes, notably of *P. protegens* subgroup strains (Fig. 4). The presence of these ISs suggests that the function associated to these TEs is conserved. ISs are well-known drivers of bacterial genome evolution through their insertions at crucial genetic locations (52, 53). They can modulate gene expression by inserting themselves in a promoter region or in the middle of a gene (54). For instance, the presence of an IS within the gene cluster involved in the lipid A formation of *Acinetobacter baumannii* provoked the complete loss of LPS production and *de facto* colistin resistance (55). Thus, as in the OBC3 gene cluster of CHA0, the IS is located upstream of the *oafA* gene, it may affect the transcription of the latter. We also identified nine genomic portions flanked by inverted repeat sequences (IRs) that are exclusively detected within the OBC3 gene cluster (Fig. 6A; Table S6). The presence of IRs has been proposed to reflect genomic instability (56). These genetic elements can form special DNA conformations such as hairpin-like or cruciform-like DNA structures that can facilitate genome rearrangements and mutations (57). IRs were detected within an LPS gene cluster of a *Xanthomonas*

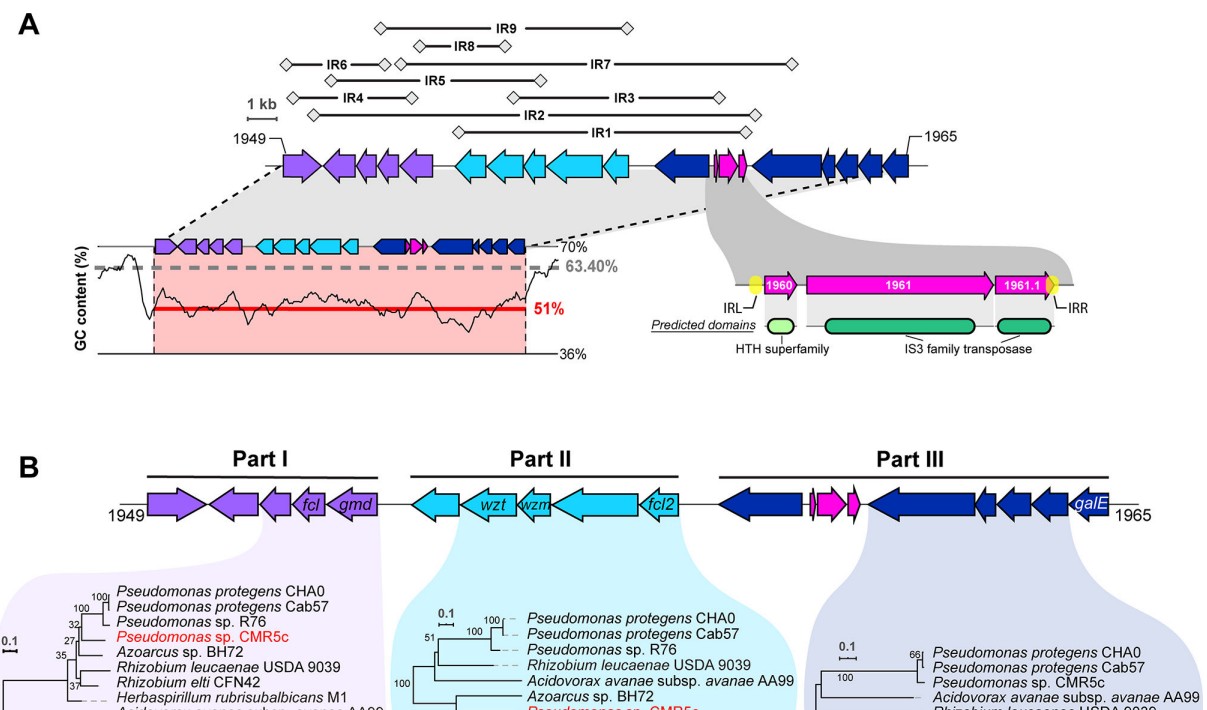

**FIG 6** The OBC3 gene cluster of *Pseudomonas protegens* CHA0 displays a high genetic plasticity and signatures of horizontal gene transfer acquisitions. (A) Nine inverted repeats (represented above the OBC3 gene cluster), which are unique at the genome scale, were detected. The nucleotide sequences of these IRs are available in Table S6. The GC content of the OBC3 gene cluster is represented at the bottom left of the gene cluster. The dark line illustrates the GC skew calculated using a 500-bp window frame. The gray dashed lines represent the average GC content of the whole genome, while the red lines correspond to the average GC content of the OBC3 gene cluster. A detailed view of the mobile genetic elements present within the OBC3 gene cluster is found at the bottom right of the gene cluster. This transposon is flanked by two inverted repeats (IRL: 5′ TGA ATC GCT CCG GGT TTC GTA GGC ACC TCT TTG CCT TAG AAT GAG GCC AA 3′ and IRR: 5′ AAG CGT TAC GCA ATG AGT TTG CAA GGT GTC TAG AGA GTC CGG GGC GAT TT 3′) and comprises two genes (PPRCHA0_1960 and PPRCHA0_1961) predicted to encode a protein containing a HTH superfamily domain and a IS3 family transposase, respectively. The third coding sequence (PPRCHA0_1961.1) corresponds to a manual annotation of a coding DNA sequence for a protein with a partial transposase domain. (B) Maximum likelihood trees were obtained from the concatenated amino acid sequences of the corresponding genes conserved in all genomes, which are highlighted in different parts. The robustness of each tree was assessed with 100 bootstrap replicates. The *Pseudomonas* strain highlighted in red displays incongruences in the tree topology compared to the species phylogenetic tree shown in Fig. 4.

strain and were proposed to either have a role in gene regulation or in recombination during horizontal gene transfer (58). Here, it could be hypothesized that the presence of IRs within the OBC3 gene cluster permitted the genomic rearrangement of the different parts of this cluster and might have facilitated their mobilization.

Parametric methods can be used to infer horizontal gene transfers by looking at genomic regions that differ from the genomic average in terms, for example, of GC content (59). CHA0 has a relatively high average GC content of 63.4%, while the GC content of the OBC3 gene cluster is 51%. Moreover, the OBC3 gene cluster is also predicted to be a genomic island by IslandViewer (60) on the basis of codon usage bias [SIGI-HMM method (61); Fig. S8]. The low GC content along with a different codon usage are sufficient to infer that the presence of the OBC3 gene cluster within the genome of CHA0 originates from horizontal gene transfer. Remarkably, most of the O-PS gene clusters investigated in this study present lower GC contents than the average of the genome in which they are located (Fig. S9). The same observation was made within other bacterial models (62). Indeed, a low GC content could be detected in all LPS clusters of *Vibrio* strains (63). The exchange and recombination of these LPS clusters led to new bacterial serogroups benefitting the adaptation of these strains to different environments (63). Low GC contents were also reported in the LPS clusters of several strains of the plant pathogen *Xanthomonas* (64). The genetic mobilization of LPS gene

clusters was also observed in *Herbaspirillum*, where clinical strains had acquired an increased number of genetic islands including LPS biosynthesis gene clusters compared to environmental strains as a possible adaptation to the human host (65). The genetic diversity of LPS in *P. aeruginosa* has also been assessed and unveiled a high heterogeneity within the O-PS biosynthesis gene clusters, which is driven by environmental selective pressures such as bacteriophages (66).

## Evolutionary origins of the OBC3 gene cluster

Phylogenetic methods are also used to evidence horizontal gene transfer based on the difference of tree topology between genes and species trees. Here, we compared the phylogenetic tree based on the deduced amino acid sequences of the OBC3 gene cluster to the species tree to identify any incongruence that would allow the inference of horizontal gene transfer. We generated different trees based on the organization of this gene cluster as mentioned above (part I, part II, and part III). Only amino acid sequences that are conserved in relevant genomes were concatenated according to the part of the OBC3 cluster they belong to, to obtain the most informative picture of the evolutionary history of this gene cluster (Fig. 6B). Different tree topologies were observed and compared to the species tree for parts I and II of the OBC3 gene cluster (Fig. 6B; Fig. S10). *Pseudomonas* sp. CMR5c, which belongs to the *P. protegens* subgroup, was found to be distant to Cab57 and CHA0 in the tree built with the amino acid sequences of part I and to be clearly distant from the other *Pseudomonas* for part II. It is likely that *Pseudomonas* sp. CMR5c could have acquired these two parts independently, contrary to other *P. protegens* strains (i.e., not by vertical transmission from a common ancestor). Nevertheless, it is evident from the tree topology that part II of the OBC3 cluster of CMR5c was not acquired from another *Pseudomonas* strain (Fig. 6B; Fig. S10). Moreover, the different genomic locations of parts I and II in the genome of CMR5c (Fig. 3) support the hypothesis of their independent acquisition compared to other *P. protegens* strains possessing the OBC3 gene cluster. On the contrary, the tree obtained from the amino acid sequences of part III is consistent with the species phylogeny. The observation that this part of the OBC3 gene cluster is located in the same genomic region in the *Pseudomonas* genomes suggests that the acquisition of part III may be older than that of the other two parts.

## Conclusion

This study provides evidence that the lytic phage ΦGP100 uses the long glycan chains of the OBC3 O-PS as a receptor for CHA0 infection. The fact (i) that several OBC3-like gene clusters could be identified in various *Pseudomonas* strains and phylogenetically more distant bacteria such as *Rhizobium* species, (ii) that the GC content differs strongly from the average GC content of the different genomes investigated in this study, and (iii) that mobile genetic markers, such as transposases, IS, IR sequences, are abundant in the OBC3 genomic region indicates that this O-PS gene cluster or parts of it have been exchanged via horizontal gene transfer. This potential genetic trade could have resulted in the high genetic and phenotypic diversity observed in the investigated bacterial LPS profiles and more generally could be a fundamental factor influencing the interactions of individuals constituting the plant microbiome with phages and presumably also with the plant host. Our results underline the complexity and the multifunctional aspect of bacterial cell surface decorations during bacterial interactions and emphasize a Cornelian dilemma of being at the mercy of predators or losing potential LPS-associated beneficial interactions. Our study provides a further illustration of how phages can be considered as a major driving force of bacterial ecology and evolution.

## MATERIALS AND METHODS

### Bacterial strains, phage, plasmids, media, and culture conditions

All strains and plasmids used in this study are listed in Tables S1, S5, and S7. Bacterial strains were routinely cultured on nutrient agar (NA), in nutrient yeast broth (NYB), or in lysogeny broth (LB) supplemented with the appropriate antibiotics when needed (ampicillin, 100 µg mL$^{-1}$; chloramphenicol, 10 µg mL$^{-1}$; kanamycin, 25 µg mL$^{-1}$; gentamycin, 10 µg mL$^{-1}$). *Pseudomonas* strains were grown at 25°C if not mentioned otherwise, while *E. coli* was cultured at 37°C. Bacteriophage ΦGP100 lysate stock was prepared as described previously (20). Briefly, the phage was added to NYB medium containing CHA0 [optical density at 600 nm (OD$_{600nm}$) = 1] and incubated overnight at 25°C. The resulting lysate was centrifuged at 2000 $g$ for 5 min and the supernatant was filtered (Millipore filters, pore size 0.22 µm). Phages were precipitated using 10% of polyethylene glycol (PEG 8000) and centrifuged (15,000 $g$, 30 min). The pellet containing phages was resuspended in SM buffer (100 mM NaCl, 8 mM MgSO$_4$·7 H$_2$O, 50 mM Tris-Cl, pH 7.5). Titration of this purified phage stock suspension (500 mL) was performed by the double layer agar technique and led to $10^9$ PFU mL$^{-1}$ (plaque forming unit per milliliter). The phage stock suspension was tested for bacterial contamination by plating 200 µL on an NA plate. Half of the phage stock suspension was kept at 4°C for up to 4 months, while the other one was aliquoted and stored in glycerol (50% vol/vol final) at −80°C for long-term storage.

### Transposon mutant library construction

A high-density Tn*5* mutant library of *P. protegens* CHA0 was prepared following a protocol described previously for the strain *P. protegens* Pf-5 (29). Briefly, competent cells of CHA0 were electroporated with the plasmid pRL27 containing the mini-Tn*5* transposon (30) and immediately rescued with 1 mL of super optimal broth with catabolite repression medium and incubated at 35°C. After 2.5 h, the obtained bacterial suspension was serially diluted and plated onto 10 NA plates supplemented with 25 µg mL$^{-1}$ of kanamycin and incubated at 25°C. Following 24 h of incubation, approximately 600,000 Km-resistant colonies were recovered and placed into a sterile 0.8% NaCl solution. The bacterial suspension was homogenized by vortexing and centrifuged to concentrate the full Tn*5*-library into a final volume of 5 mL of which aliquots of 1 mL were stored with glycerol (50% vol/vol final) at −80°C for subsequent use.

### Selection of phage-resistant Tn*5*-mutants and sequencing

Aliquots of 150 µL of the Tn*5*-mutant library were added to 8 mL of NYB medium and incubated for 8 h at 25°C. The OD$_{600nm}$ of this culture was adjusted to 0.01 (2 × $10^6$ cells mL$^{-1}$) in four tubes containing 10 mL of fresh NYB. Three of these tubes were inoculated with ΦGP100 phage stock solution at different concentrations; $10^6$, $10^7$, and $10^8$ PFU mL$^{-1}$ (i.e., at an MOI of 1, 10, and 100, respectively). The last tube was inoculated with SM buffer as a control. The four tubes were then incubated at 25°C for 10 h (corresponding to approximately 3,000 bacterial generations). After incubation, the cells were centrifuged and washed three times with sterile water to remove any traces of DNA from lysed cells. Genomic DNA was then extracted from the cell pellets using the MagAttract HMW DNA kit (Qiagen). The library preparations and paired-end sequencing (HiSeq2500, Illumina) were performed as detailed previously (29). Three independent biological replicates were performed.

### Sequence processing and statistical analysis

The reads were processed and analyzed as previously described (29). Reads were preprocessed using cutadapt [v.2.3 (67)] and reaper [v.15−065 (68)] and mapped to the genome of *Pseudomonas protegens* CHA0 (LS999205.1) using bwa [v.0.7.17 (69)]. FeatureCounts (v.1.6.0) was used to summarize the number of read counts per gene

locus and TRANSIT (v.2.5.2) was used for statistical analysis. Insert site counts were normalized using the TTR method (70). The comparative analysis for determining the conditional essentiality of genes was performed using the resampling method [10,000 permutations (70)]. We considered bacterial genes enriched following the phage exposure when the adjusted $P$ value < 0.05 and $\log_2$FC > 2. At the opposite, negatively selected genes were assigned when $P$ value < 0.05 and $\log_2$FC < −2.

## Genomic analysis

Functional annotation of genes into clusters of orthologous groups (COGs) from the *P. protegens* CHA0 genome was performed using eggNOG-mapper (v2) (71). The synteny and the detection of sequence homology of the OBC3 gene cluster were obtained by cblaster (72, 73) with the homology threshold set at 60% of amino acid sequence identity on at least 70% of the sequence length. Inverted repeats in the OBC3 gene cluster of CHA0 were detected by GeneQuest (DNASTAR's software, v15.3). The GC contents were calculated with a window range of 500 bp using Artemis (74). The functional annotation of the phage ΦGP100 was updated using PROKKA (75) with a priority annotation using the PHASTER protein database [version 22 December 2020 (76)]. The similarity between the different phage genomes was calculated and visualized using Circoletto (77). The protein structure of the pectate lyase was predicted using AlphaFold2 (78). Protein structure comparison was performed using the Dali server (79) and the protein domain was detected by interrogating the Conserved Domain Database from NCBI (80).

## Phylogenies

The maximum-likelihood (ML) phylogenetic tree of the 34 bacterial genomes (Table S5) was built based on single-copy proteins, as previously described (81). Briefly, proteomes were analyzed with OrthoFinder v2.2.6 (82) using diamond v0.9.21.122 (83) searches. Orthologous sequences of 478 single-copy proteins present in all the genomes were further aligned with Clustal Omega (84) and concatenated. Gblocks v0.91 (85) was used to remove highly divergent regions and poorly aligned columns. The resulting alignment of concatenated sequences was imported into RAxML-NG v1.0.2 (86) to build the ML phylogeny. The LG model of amino acid evolution (87), gamma-distributed substitution rates, and empirical frequencies of amino acids were used. Fast bootstrap was applied, with a subsequent search for the best-scoring tree (88) and autoMRE (89). For the phylogenetic reconstruction of the different parts of the OBC3 cluster, amino acid sequences were retrieved and aligned using Clustal Omega. Maximum likelihood trees were obtained from these aligned sequences using the LG substitution model (87) with SPR topology search. The robustness of each tree was assessed with 100 bootstrap replicates. The taxonomy of the phage ΦGP100 was assigned according to the proteomic tree generated on the online platform ViPTree (46).

## Mutant construction

Different mutants of *P. protegens* CHA0 were constructed (Table S7) to confirm the involvement of specific genes in phage resistance following the results of the Tn-seq experiment. These mutants were obtained using the suicide vector pEMG and the I-SceI system (90) with a protocol adapted for *P. protegens* CHA0 (29, 90), with plasmids and primers listed in Tables S1 and S8, respectively.

## Heterologous expression of *fcl* from *Rhizobium etli* CFN42 and lipopolysaccharide visualization

For complementation of the Δ*fcl* mutant of CHA0 *in cis* with the *fcl* gene of *Rhizobium etli* CFN42 or *in trans* with the *fcl* gene of CHA0, the respective *fcl* genes were cloned under the control of the P*tac/lacIq* promoter in the mini-Tn*7* delivery vector pME8300 (91), generating the pME11021 and pME11022, respectively (Table S1). The pME8300 derivatives and the Tn*7* transposition helper plasmid pUX-BF13 were co-electroporated

into competent cells of the CHA0 Δ*fcl* mutant to create the complemented strains CHA5211 and CHA5212, having $P_{tac/lacIq}$-*fcl*(CFN42) or $P_{tac/lacIq}$-*fcl*(CHA0), respectively, integrated at the unique chromosomal Tn*7* attachment site (Table S7). Expression of *fcl in cis* or *in trans* was then induced by adding isopropyl β-D-1-thiogalactopyranoside (IPTG) at a final concentration of 0.1 mM to the cultures. Extraction, separation and visualization of LPS were performed as previously described (28).

## Phage sensitivity assay

Infection of different *Pseudomonas* strains and isogenic mutants of CHA0 by the phage ΦGP100 was assessed by a double agar layer assay. Aliquots of 4 mL of LB soft agar (5 g L$^{-1}$ oxoid bacteriological agar), supplemented with IPTG if relevant, were mixed with 100 µL of cell suspension of a given strain that was prepared from an overnight culture grown in 10 mL of LB (supplemented with antibiotics and IPTG, if relevant) at 25°C with shaking at 180 rpm. The mixture then was poured onto a sterile NA plate. The phage stock suspension (containing 10$^9$ PFU mL$^{-1}$) was serially diluted in a 96-well plate and 10 µL of each dilution was spotted onto the solidified double layer. Plates were evaluated for the occurrence of plaques in the bacterial growth after overnight incubation at room temperature.

## Phage burst size and adsorption rate characterization

The adsorption rate and the number of new phages released from infected cells, i.e., the burst size of the phage ΦGP100, were determined following the infection of the different CHA0 mutants as compared to the parental strain. A bacterial suspension of 900 µL containing $2.5 \times 10^8$ bacterial cells at exponential growth stage (OD$_{600nm}$ = 1) was supplemented with 100 µL containing $2.5 \times 10^7$ PFU, i.e., at an MOI of 0.1, and incubated at room temperature for 5 min. A total of 100 µL of this suspension was collected in a new tube containing 5 µL of chloroform to determine the adsorption rate. The rest of the suspension was diluted to obtain a suspension containing 10$^5$ bacterial cells in 10 mL of fresh NYB. The burst size was monitored by sampling 100 µL at 60 min then every 10 min until 150 min. Final samples were collected at 180 min. The collected samples were placed into a new tube containing 5 µL of chloroform, incubated for 10 min, and then caps of the tubes were opened 10 min before the samples were diluted into a 96-well plate. The enumeration of phages was then performed by spotting 10 µL of each dilution onto the solidified double layer containing CHA0 wild type.

## ACKNOWLEDGMENTS

We thank Dale Noel from Marquette University for providing us with *Rhizobium etli* CFN42 and Laure Weisskopf from the University of Fribourg for providing us with *Pseudomonas* sp. R76.

This study was supported by the Swiss National Centre of Competence in Research (NCCR) Microbiomes (no. 51NF40_180575) and 310030_184666 from the Swiss National Science Foundation (SNSF). Open access funding was provided by the University of Lausanne.

## AUTHOR AFFILIATIONS

[1]Department of Fundamental Microbiology, University of Lausanne, Lausanne, Switzerland
[2]Plant-Microbe Interactions, Department of Biology, Science4Life, Utrecht University, Utrecht, the Netherlands

## PRESENT ADDRESS

Peter Kupferschmied, Federal Office for Agriculture, Swiss Federal Plant Protection Service, Bern, Switzerland

## AUTHOR ORCIDs

Jordan Vacheron http://orcid.org/0000-0003-0031-1338
Clara M. Heiman http://orcid.org/0000-0003-4550-7537
Julian R. Garneau http://orcid.org/0000-0003-4579-8776
Ronnie de Jonge https://orcid.org/0000-0001-5065-8538
Daniel Garrido-Sanz https://orcid.org/0000-0003-3279-6421
Christoph Keel http://orcid.org/0000-0002-8968-735X

## FUNDING

| Funder | Grant(s) | Author(s) |
|---|---|---|
| National Centre of Competence in Research Microbiomes | 51NF40_180575 | Daniel Garrido-Sanz |
| | | Julian R. Garneau |
| | | Christoph Keel |
| | | Jordan Vacheron |
| Swiss National Science Foundation | 310030_184666 | Clara Margot Heiman |
| | | Christoph Keel |

## AUTHOR CONTRIBUTIONS

Jordan Vacheron, Conceptualization, Data curation, Formal analysis, Investigation, Methodology, Project administration, Resources, Software, Supervision, Validation, Visualization, Writing – original draft, Writing – review and editing | Clara M. Heiman, Investigation, Writing – review and editing | Julian R. Garneau, Software, Writing – review and editing | Peter Kupferschmied, Conceptualization, Investigation, Methodology | Ronnie de Jonge, Writing – review and editing | Daniel Garrido-Sanz, Resources, Writing – review and editing | Christoph Keel, Conceptualization, Funding acquisition, Project administration, Supervision, Writing – review and editing

## DATA AVAILABILITY

The generated Tn-seq datasets were deposited on the EBI platform with sample accession numbers SAMEA7616892 to SAMEA7616904 under the BioProject PRJEB61467.

## ADDITIONAL FILES

The following material is available online.

### Supplemental Material

**Dataset 1 (Spectrum02049-23-s0001.xlsx).** Contain the data obtained from the Tn-Seq experiment.
**Supplementary information (Spectrum02049-23-s0002.pdf).** Contain all supplementary figures and tables.

### Open Peer Review

**PEER REVIEW HISTORY (review-history.pdf).** An accounting of the reviewer comments and feedback.

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
