## [Reviewer comments · Microbiology Spectrum]

Microbiology Spectrum

Molecular and evolutionary basis of O-antigenic polysaccharide driven phage sensitivity in environmental pseudomonads

Jordan Vacheron, Clara Heiman, Julian Garneau, Peter Kupferschmied, Ronnie de Jonge, Daniel Garrido-Sanz, and Christoph Keel

Corresponding Author(s): Jordan Vacheron, Universite de Lausanne

Review Timeline:

Submission Date:	May 16, 2023
Editorial Decision:	August 9, 2023
Revision Received:	August 15, 2023
Accepted:	August 16, 2023

Editor: Eric Cascales

Reviewer(s): Disclosure of reviewer identity is with reference to reviewer comments included in decision letter(s). The following individuals involved in review of your submission have agreed to reveal their identity: Leo Eberl (Reviewer #1); Sanghamitra Saha (Reviewer #2)

Transaction Report:

DOI: <https://doi.org/10.1128/spectrum.02049-23>

August 9, 2023

Dr. Jordan Vacheron
Universite de Lausanne
Department of Fundamental Microbiology
Biophore Building
Quartier UNIL-Sorge
Lausanne, Vaud 1015
Switzerland

Re: Spectrum02049-23 (Molecular and evolutionary basis of O-antigenic polysaccharide driven phage sensitivity in environmental pseudomonads)

Dear Jordan:

Thank you for submitting your manuscript to Microbiology Spectrum. Again, I wish to apologize for the overly long delay of this reviewing, but as we already discussed, we had difficulties to secure two reviewers. I have now received the comments from two reviewers, and as you will see (pasted below), both reviewers acknowledge that you report new findings, although not unexpected, and that your manuscript is very clearly presented and written. They both recommend publication with very minor changes. I therefore encourage you to respond to reviewer #1 and invite you to submit your revised manuscript. When submitting the revised version of your paper, please provide (1) point-by-point responses to the issues raised by the reviewers as file type "Response to Reviewers," not in your cover letter, and (2) a PDF file that indicates the changes from the original submission (by highlighting or underlining the changes) as file type "Marked Up Manuscript - For Review Only". Please use this link to submit your revised manuscript. Detailed instructions on submitting your revised paper are below.

Link Not Available

Sincerely,
Eric

Eric Cascales

Reviewer comments:

Reviewer #1 (Comments for the Author):

This study identified molecular determinants in the biocontrol agent *Pseudomonas protegens* CHA0 that confer sensitivity to phage Φ GP100 employing a transposon-sequencing (Tn-Seq) approach. The use of Tn-Seq to identify factors involved in phages sensitivity by identifying positively selected mutants in the presence of the phage is a refreshingly new experimental approach. While it would not change the main results of this study I wonder whether number of genes detected as potentially involved in the bacterial sensitivity is a bit high (about 300) and may reflect a relatively long exposure time (10 h, how many generations?) or that the threshold set is too low (I may have overlooked it, but I could not find the criteria used to judge whether a gene is considered important or not). If the threshold is increased the role of LPS biogenesis may become even more pronounced. Did the authors also observe negatively selected genes that would be worthwhile to mention?

The authors also performed a number of additional experiments to prove the results of the global Tn-Seq analysis and provide clear evidence that LPS biogenesis is key for Φ GP100. While this is not unexpected, I think this is a very complete and convincing study that is extremely well designed, executed and presented. The data and discussion on the evolutionary aspects of LPS and phage resistance were a pleasure to read.

Reviewer #2 (Comments for the Author):

Pseudomonas sp CHAO is a biocontrol agent in preventing damage by lepidopteran pests and to do it has to colonize the plant roots. Phages such as 0GP100 can prevent this colonization by lysing the *pseudomonas* species. Certain markers on the LPS namely OBC-3 type O antigenic polysaccharide allows attachment of the phage. The authors were interested in finding out the genes responsible for phage resistance by using a Tn library approach and identified genes responsible for attachment. Further studies were also carried out by deletion studies highlighting the importance of the OBC 3 Type O LPS. The role of the genes involved in infection was tested by plaque assays.

The distribution of the *obc3* gene cluster was checked and shown to be present in other bacterial species. Phylogenetic analysis showed that this gene cluster could have been obtained through horizontal gene transfer.

Preparing Revision Guidelines

Please return the manuscript within 60 days; if you cannot complete the modification within this time period, please contact me. If you do not wish to modify the manuscript and prefer to submit it to another journal, please notify me of your decision immediately so that the manuscript may be formally withdrawn from consideration by Microbiology Spectrum.

Title: Molecular and evolutionary basis of O-antigenic polysaccharide 2 driven phage sensitivity in environmental pseudomonads

Comments to the authors:

Pseudomonas sp CHAO is a biocontrol agent in preventing damage by lepidopteran pests and to do it has to colonize the plant roots. Phages such as OGP100 can prevent this colonization by lysing the *pseudomonas* species. Certain markers on the LPS namely OBC-3 type O antigenic polysaccharide allow attachment of the phage. The authors were interested in finding out the genes responsible for phage resistance by using a Tn library approach and identified genes responsible for attachment. Further studies were also carried out by deletion studies highlighting the importance of the OBC 3 Type O LPS. The role of the genes involved in infection was tested by plaque assays.

The distribution of the *obc3* gene cluster was checked and shown to be present in other bacterial species. Phylogenetic analysis showed that this gene cluster could have been obtained through horizontal gene transfer.

The paper is very comprehensive and detailed, and the authors have provided different lines of evidence showing the role of the long O-PS for the infection by OGP100 phage. The role of the depolymerase gene of the phage was shown by the formation of turbid plaques in the plaque assay using CHAO as the host. The phage appears to belong to the Zobellviridae family based on genomic analysis of the phage.

Point-by-point response to the referees

Manuscript number: Spectrum02049-23

Title: "***Molecular and evolutionary basis of O-antigenic polysaccharide driven phage sensitivity in environmental pseudomonads***"

The line numbers we refer to along this file correspond to the line numbers found in the Marked-Up Manuscript (PDF) file.

Reviewer #1 (Comments for the Author):

This study identified molecular determinants in the biocontrol agent *Pseudomonas protegens* CHA0 that confer sensitivity to phage Φ GP100 employing a transposon-sequencing (Tn-Seq) approach. The use of Tn-Seq to identify factors involved in phages sensitivity by identifying positively selected mutants in the presence of the phage is a refreshingly new experimental approach. While it would not change the main results of this study I wonder whether number of genes detected as potentially involved in the bacterial sensitivity is a bit high (about 300) and may reflect a relatively long exposure time (10 h, how many generations?) or that the threshold set is too low (I may have overlooked it, but I could not find the criteria used to judge whether a gene is considered important or not). If the threshold is increased the role of LPS biogenesis may become even more pronounced.

>> We appreciate the comments provided by reviewer #1. Reviewer #1 made a valid point about the lengthy exposure time (10 hours, which equals to approximately 3,000 bacterial generations - We have now added this information to line 395 of the revised manuscript). The exposure time also accounts for the similarities in Tn-seq results across different phage concentrations. This is now stated in lines 141-142.

To determine the importance of a gene, we applied a threshold based on log2 fold change and the P value resulting from comparing gene count in the presence and absence of phages. While this information is available in the related-figure captions, we appreciate the reviewer bringing it to our attention and have included it in the Materials and Methods section on line 408-410.

Did the authors also observe negatively selected genes that would be worthwhile to mention?

>> We appreciate that the reviewer #1 raised this question. Certainly, the insertion of the transposon could result in bacterial mutants that are highly sensitive to the phage, more than the wild-type strain. In this scenario, genes that were disturbed by the transposon insertion are subjected to negative selection throughout the experiment.

We identified 3,476, 4,029, and 4,074 genes that were significantly less represented in the Tn-mutant library after the phage exposure, based on the applied threshold (Fig. 1 and Fig. S1). We manually screened for genes related to membrane biosynthesis or DNA processing but found that this set of genes consisted of essential ones required for the development and growth of the bacterial host. If certain bacterial genes that hinder the phage infection process are selectively disadvantaged, they are most probably integrated within all these essential genes identified regarding this experimental set-up. That is why we chose not to elaborate further on this, and to focus on the cell surface decorations.

The authors also performed a number of additional experiments to prove the results of the global Tn-Seq analysis and provide clear evidence that LPS biogenesis is key for Φ GP100. While this is not unexpected, I think this is a very complete and convincing study that is extremely well designed,

executed and presented. The data and discussion on the evolutionary aspects of LPS and phage resistance were a pleasure to read.

>> We thank reviewer #1 for all the comments and questions which greatly contributed to improving our manuscript.

Reviewer #2 (Comments for the Author):

Pseudomonas sp CHAO is a biocontrol agent in preventing damage by lepidopteran pests and to do it has to colonize the plant roots. Phages such as OGP100 can prevent this colonization by lysing the *pseudomonas* species. Certain markers on the LPS namely OBC-3 type O antigenic polysaccharide allows attachment of the phage. The authors were interested in finding out the genes responsible for phage resistance by using a Tn library approach and identified genes responsible for attachment. Further studies were also carried out by deletion studies highlighting the importance of the OBC 3 Type O LPS. The role of the genes involved in infection was tested by plaque assays. The distribution of the *obc3* gene cluster was checked and shown to be present in other bacterial species. Phylogenetic analysis showed that this gene cluster could have been obtained through horizontal gene transfer.

>> We thank reviewer #2 for this accurate summary appreciating our manuscript.

August 16, 2023

Dr. Jordan Vacheron
Universite de Lausanne
Department of Fundamental Microbiology
Biophore Building
Quartier UNIL-Sorge
Lausanne, Vaud 1015
Switzerland

Re: Spectrum02049-23R1 (Molecular and evolutionary basis of O-antigenic polysaccharide driven phage sensitivity in environmental pseudomonads)

Dear Jordan, dear Christoph,

Thank you for submitting your revised manuscript. I am pleased to accept it for publication in Microbiology Spectrum. You will be notified when your proofs are ready to be viewed.

Sincerely,
Eric

Eric Cascales
Editor, Microbiology Spectrum
